# Nurses' perspectives on their communication with patients in busy oncology wards: A qualitative study

**E. Angela Chan** *, **Pak Lik Tsang**, **Shirley Siu Yin Ching**, **F. Y. Wong**,
**Winsome Lam**

School of Nursing, The Hong Kong Polytechnic University, Hung Hom, Kowloon, HKSAR

☯ These authors contributed equally to this work.
\* e.angela.chan@polyu.edu.hk

**Data Availability Statement:** All relevant data are withing the manuscript and its supporting information files.

## Abstract

### Background

Despite an increase in emphasis on psychosocial care in cancer nursing, time constraints and nurses' lack of knowledge in skilled communication continue to be challenges.

### Aims

To examine how cancer care nurses view their communication with patients and how they deal with the psychosocial needs of patients in busy wards.

### Design

A qualitative interview study.

### Methods

Focus groups and individual interviews were conducted with eleven hospital-based cancer nurses in Hong Kong from July 2, 2017 to January 2, 2018.

### Results

A qualitative thematic analysis of the data identified three themes: 1. Intentional and unintentional psychosocial care that is secondary in focus; 2. Managing an emotionally challenged environment; 3. Mentoring and learning.

### Conclusion

Oncology settings are time-constrained, emotionally charged environments for nurses, and providing psychosocial care for patients is a secondary concern. While proactive strategies can be used to avert patient complaints, being open and attending to the individual needs of patients is equally important to avoid blocking in nurse-patient communication. Despite emotional entanglement and tensions, the positive follow-up strategies used by nurses to manage the patients' emotions and provide psychosocial care reflect good practices.

**Funding:** This study was funded by the General Research Committee (no. 156003/15H) in Hong Kong. The funder had no role in study design, data collectin and anlysis, decision to publish and preparation of the manuscript.

**Competing interests:** The authors have declared that no competing interest exist.

Leadership and support are needed to deal with the nurses' perception that their communication training has been ineffective and their ability to manage strong emotions deficient. Communication skills, honed by making continuous opportunities to communicate available, as well as an understanding of emotional labour, need to be integrated with mindfulness in the nurses' care of themselves and their patients. Notwithstanding the importance of experience in oncology care for junior nurses, it is necessary for both junior and senior nurses to learn about and reflect upon the different forms of emotional labour if value-based care is to be provided. In addition, it is essential for junior nurses to receive continuous coaching and mentoring, and to engage in reflective learning from each clinical encounter with oncology patients.

## Introduction

Nurses involved in cancer care are facing ever greater demands and must deliver care more quickly for economic reasons and because of a worldwide shortage of nurses [1]. The Hospital Authority, as the largest healthcare provider in Hong Kong, is facing the problem of a major understaffing of nurses in all settings [2]. The nurse-patient ratio in a ward is around 1:11/12 in Hong Kong, while the international standard is 1:4–6 [3]. Aiken, Sloane [4] asserted that international studies have revealed that nurse shortages have substantial implications for patient care. These include high rates of patient mortality, infection, drug errors and accidents, and longer stays in hospital [5]. West, Mays [6] asserted that nursing workloads and time pressures hinder nurses from attending to the fears and concerns of patients and giving them comprehensive information. Nurses tend to accord the highest priority to required tasks with immediate and visible effects, and the lowest to emotional care [7].

Notwithstanding the issue of time constraints, Engel [8] proposed that the biomedical model be replaced by a holistic and multidimensional approach addressing complex bio- psycho-social interactions for the well-being of patients. He argued that the biopsychosocial model will bridge the separation between mind and body. Providing holistic care under this model will help to meet the physical and psychosocial needs of patients. Currently, many nurses are educated under a biomedical model [9], with the result that the care that they provide often focuses only on the physical aspect of a patient's needs.

Pehrson, Banerjee [10] asserted that cancer patients have a great need for information and emotional support. However, as these patients do not always disclose their concerns directly to nurses, nurses need to be able to recognize their verbal and non-verbal cues. An overlooked psychological need may lead to distress if prompt attention is not provided [11]. Hogg, Hanley [12] also concluded that the sensitivities of many patients and carers were heightened when faced with serious health conditions such as cancer and health-related stresses, and that they expected health professionals to be compassionate and kind. In addition, there are complex contextual factors that affect the sensitivity of patients and carers to staff attitudes, behaviour, and communication. The failure of health professionals to meet the norms of communication expected by patients during their encounters with them will lead to complaints from patients and significant negative effects on the patients' feelings and subsequent healthcare experiences. Hence, the importance of adequate and effective nurse-patient communication cannot be overemphasized.

While there are numerous evidence-based courses on communication skills to help healthcare professionals interact more effectively with patients [13], an earlier study by Wilkinson

[14] has already supported the notion that when and how nurses communicate may depend on their work environment and on their beliefs and attitudes towards death and dying, rather than on specific training in communication. Salmon and Young [15] concurred that training in communication skills does not necessarily turn nurses into skilled communicators. As all clinical situations differ to some extent, communicating with patients inherently requires creativity and flexibility.

Despite the focus on improving 'communication skills' and on determining the optimal timing for nurse-patient communication to occur as part of holistic care [16], relatively little is known about how nurses articulate their experiences of communicating with cancer patients in a busy environment, especially with regard to psychosocial issues. In the recent literature on psycho-oncology, emphasis has been placed on the integration of psychosocial aspects into the care of patients with cancer [7]. However, several studies have shown that nurses tend to focus more on the physical than on the psychosocial needs of cancer patients, and that the latter is often not considered part of the routine practice of nursing [17, 18]. Aldaz, Treharne [19] also reported that healthcare professionals seldom engage cancer patients in discussions about the emotional uncertainties that they might be facing, hoping instead that the patients will receive support from spiritual services. On the other hand, Chan, Wong [20] recently found that what cancer patients expect from nurses is effective physical care, given their understanding of the nurses' heavy workload. Similarly, Brown, Lui [21], in an Australian study, found that patients preferred to seek emotional support from close family members or to manage on their own. The intention here is not to dismiss the significance of the psychosocial needs of oncology patients, as unattended psychosocial issues can lead to distressing mental problems [22]. We are currently aware of only one study [23] on nurses' perceptions and their strategies for providing oncology care in a time-strapped context. Not only was the quality of care significantly compromised in such an environment, but the nurses also felt an extra emotional burden as they struggled to provide quality care to the patients [23]. Acknowledging the perspectives of nurses regarding their perceptions of how they communicate with cancer patients and respond to patients' psychosocial needs in a time-strapped environment will shed light on how to improve the quality of oncology care.

## Materials and methods

### Design

This is a qualitative interview study, which forms part of a larger, focused ethnographic study on nurse-patient communication in the oncology wards ($n = 2$) of an acute hospital. The qualitative interview is a suitable way to solicit views and to collect detailed accounts of participants' thoughts, attitudes, beliefs, and knowledge pertaining to a given phenomenon [24]. Carey [25] suggested that focus group interviews, in particular, are 'especially well-suited for problems in health research where complex clinical issues (such as nurse-patient communication in oncology care) are often explored through qualitative research' (p. 227).

### Aims

To examine how cancer care nurses viewed their communication with patients and how they dealt with the psychosocial needs of patients in busy wards.

### Participants

This study involved 11 interviews with nurses. These nurses had participated in the larger focused ethnographic study on addressing patients' views of nurse-patient communication in

the same oncology wards. They were purposively sampled and recruited by the research assistant (RA). The majority of the participants were female (90.9%) and had a bachelor's degree in nursing (63.6%). Nurses who had been working in oncology between 3 to 5 years and those who had worked for over 5 years constituted 45% of each group (S2 Table).

There was no pre-existing relationship between the nurses and the researchers. The criteria for the inclusion of nurses were those with at least two years of nursing experience and one year in the current oncology work setting.

## Data collection

Taking into consideration the nurses' busy schedules, they were asked to choose the most convenient way for them to be interviewed. Individual and focus group interviews were conducted from July 2, 2017 to January 2, 2018. The result was that two focus group interviews, which consisted of four to five participants per group, were conducted by a skilled moderator and a note taker, who were the researchers. This was interspersed by two individual interviews that were conducted by the RA. The individual interviews lasted for 45 minutes, while the focus group interviews lasted for 60 to 90 minutes. Interviews were conducted until thematic data saturation was reached after the interviews with 11 participants, where no new/additional ideas were identified in the data [26]. A sample size of 11 is also typically be enough to reach data saturation [27].

The team tried to reconcile the different accounts elicited from the nurses according to the method (whether individual or focus group) that was used to interview them. Coenen, Stamm [28] suggested using a common reference to meet the challenge of comparing results. In our case, guided questions were used to compare the data.

The interview guide (S1 and S2 Files) was developed from the literature. The set of guided questions started from the general and moved on to the specific in addressing nurse-patient communication for oncology care in a busy environment. The findings from the first part of the study [20], on the patients' perceptions of their nurse-patient communication in a busy ward [20] (S1 Table), were used to stimulate discussion with the nurses after they were asked some general questions about their thoughts on their work, before specific questions were raised about communication. Nurses were encouraged to reflect on the patients' sharing of their views and their ways of looking at nurse-patient communication and psychosocial care.

## Researcher's reflexivity

According to Dowling [29], reflexivity in the qualitative research process can take on forms such as personal reflexivity and epistemological reflexivity. In the case of personal reflexivity, each member of the research team often found herself/himself ruminating on how her/his values, beliefs, and experiences might shape the discussions. We also pondered the question of how the findings might have affected us as educators, professionals, and persons. The following examples are illustrative.

When we were reading the transcripts about the nurses' reactions to the comments of the patients from their wards regarding the nurses' tone, choice of words, and the subsequent non-verbal walking away behaviour, the research team tended to make judgements based on our value system about the nurses' choice of words. After the discussion of our need to be more open to the justification of the senior nurse that the characteristics of the patient warranted the firm verbal behaviour that she would employ to capture the patient's attention, we also became more respectful of the experience and sympathetic about her comment. Despite this understanding, in the larger study [20] it was clear, from the perspective of the patient, that the nurse had failed to provide effective communication, especially when she walked away

from the patient. From the feedback provided by the patient, it could be discerned that the relationship between them had been damaged.

For the discussion among the research team focusing on the nurse's use of proactive strategies to avert complaints, it initially has led to reflections on the importance and the value of practice-based theory. However, it was necessary to consider that the nurse's predetermined understanding might undermine the patient's individualized needs. This eventually caused us to think hard about the need to strike a balance between predetermined understandings and individual needs.

On the issue of epistemological reflexivity, we were faced with a dilemma relating to our methodology. As other researchers have done, we combined both methods for practical considerations, that is individual interviews were offered to accommodate the busy schedules of those participants who were unable to attend a focus group interview [30, 31]. However, as we proceeded to analyse the data, we followed the work of Lambert and Loiselle [24] on combining the data from the individual and focus group interviews. This gave us a richer and more nuanced understanding of the phenomenon than we initially anticipated. We then focused on investigating how the combination of methods enhanced our understanding of the patterns of nurse-patient communication in busy oncology settings. When comparing the transcripts from the focus group and individual interviews, two levels of understanding of the phenomenon were noted.

Data from the focus groups revealed a general understanding of the range of patterns of nurse-patient communication; for instance, psychosocial care can be delivered as planned in routine care and on an ad hoc basis when following up if time permits. The individual interviews, on the other hand, offered detailed descriptions of how individuals proceeded through a particular pattern of communication, and further enriched the initial conceptualisation of the phenomenon. Thus, the separate data sets were mutually informative. During the individual interviews, the researcher used the initial understanding from the first focus group as a guide to discern whether or not and how an individual nurse's experience of nurse-patient communication could be embedded in the overall pattern. In this way, individual patterns were not explored in isolation. For instance, the individual interview may allow more personal experiences to be shared, such as one nurse's description of how she learned to manage the intense distress of a patient's wife through the use of touch as non-verbal communication, and by reminding her that the children would still be with her, even though her husband (the patient) had passed away.

Focus groups on the other hand facilitated an exploration of opinions and beliefs about the phenomenon of nurse-patient communication. For instance, the nurses stated that, with some patients, the issue might be an emotional one, and in that case, they would show empathy when managing the patient, but others stated that the issue could be about the characteristics of the patient. Therefore, different nurses would manage the situation and behave differently in terms of using a firm tone to convey their message to the patient.

In turn, the focus group data makes it possible to view and interpret the data obtained from the personal interviews through a wider lens. The interviews add depth to show that the contexts of care are constantly changing and that what is important is to be flexible and creative. This also leads to an understanding of the importance of experiential learning.

The above view is also supported by the Zoom Model [32], which identifies different levels of meaning in the participants' narration of a phenomenon. For example, macro-zoom would be the focus group data (corresponding to the socio-historical dimension and collective meanings), whereas the interview data would be the meso-zoom (reflecting a personal level of values). Hennink, Kaiser [33] concluded that focus group interviews generated more information as a result of the interactions of the group, providing more breadth (i.e., a contextualised

understanding of nurse-patient communication in an emotionally charged oncology environment), while the individual interviews added more depth. Moezzi [34] also found that focus groups were particularly useful at cataloguing the range of the participants' experiences, while individual interviews provided details of those experiences. Given this understanding, a comparison of the data led to an identification and interpretation of the individual and contextual circumstances surrounding the phenomenon of nurse-patient communication, and to an understanding of emotional labour and the need for training.

## Data analysis

Having been actively involved in collecting the data, the researchers were familiar with the data before they conducted the analysis. In the case of the focus groups, the moderator and the recorder held a debriefing session after each group interview, to review their impressions of the interview and the recorder's notes. After the individual interviews, the RA discussed the interviews with the first author. Both then immersed themselves further in the data. Braun and Clarke [35] six-step framework for a thematic analysis was adopted, with the write-up as the final step. Since it is a method rather than a methodology, it is not tied to a particular theoretical perspective; thus, it provides more flexibility than a theory [36]. Since our research question reflects the researchers' interest in the oncology nurses' own accounts of how they viewed their communication with patients and how they dealt with the psychosocial needs of patients in busy wards, this question determined the direction of the interview questions and the analysis. The research team also allowed the process to be driven by the data when issues were raised by the participants, which gave the process a more inductive focus. This was the case even though our analysis was primarily driven by specific research questions, which should have made the exercise more of a theoretical thematic analysis.

Our first task was to read and re-read the data from the whole data set, to familiarise ourselves with the data before commencing the job of coding. At this stage, the research team took notes and jotted down their early impressions. The process of generating initial codes started with two independent coders (the first author and the RA) looking for patterns of meaning and issues relating to nurse-patient communication and psychosocial care in oncology settings. Each segment of the data was coded in accordance with its relevance or if it captured something interesting about the research question.

Coding was conducted manually by writing notes as keywords, phrases, and sentences on the transcripts (e.g., nurses' psychosocial care, patients' emotions, and mentoring). The nurses' responses in the individual and focus group interviews were colour coded to match them with those of the individual respondents. Comparisons were made between the data, with new codes being recorded as they were identified. The third author compared the codes for consistent usage and rectified discrepancies before the whole data set was coded. The analysis of the data was carried out concurrently with the collecting of the data, through constant comparisons [37]. The team then searched for themes. Significant codes were those that were cited frequently and formed a coherent pattern. Data were scrutinized for recurrent patterns of similar and different meanings. In reviewing the themes, it was found that a preliminary theme (responding to patients' cues) did not have much data to support it and that it could be embedded into a revised theme of psychosocial care as secondary. There was also an initial expectation that the expected care in oncology would be a preliminary theme, but it was eliminated. Themes were reviewed with all of the collected data relevant to each theme arranged into subthemes and combined to form potential overarching themes. In defining the themes, some that were found to be significant across the data were used to develop the final thematic map (S1, S2 and S3 Figs). These included expected communication on psychosocial care and its

priority; contextual influence and environment on emotions and communication management; and nurses' experience and support through training, mentoring, and continuous learning. A thematic map was created to show the connecting subthemes and the main themes [35]. Agreement on the three major themes was reached through extensive iterative discussions among the members of the research team at the regular meetings. The process was completed with a written analysis that provided sufficient evidence of the themes within the data [35].

### Rigour

To ensure trustworthiness, the transcripts were checked against the audio recordings. For dependability, fieldnotes were also taken to better understand the interactions that occurred during the group sessions. For credibility, meetings were held to reach agreement with a third reviewer, who evaluated the coding for consistency and new themes.

### Ethical considerations

Approval was obtained from the Research Ethics Committee of the hospital in the Kowloon West Cluster (REC no. 87–14) and the Departmental Research Committee of the Hong Kong Polytechnic University (HSEARS 20130924002–01). Consent was received from the nurses after an explanation was given to the ward manager and nurses of the intent of the research (i.e., to understand how best to support nursing care and nurse-patient communication in busy oncology settings). The participants were told that pseudonyms would be used for anonymity. They were also informed that no consequences to them would result from discussing any of the issues in the study or from withdrawing from the study.

## Results

A qualitative thematic analysis of the data identified three themes: 1. Intentional and unintentional psychosocial care that is secondary in focus; 2. Managing an emotionally challenged environment; and 3. Mentoring and learning. The different interviewees were identified with different alphabet codes. The abbreviations F and I were assigned to designate focus group interviews and individual interviews, respectively. Data extracts from focus groups 1 and 2 were indicated using (F1) and (F2), respectively.

### Intentional and unintentional psychosocial care that is secondary in focus

The nurses' accounts of how they prioritised their work revealed that they would attend to the physical needs of the patient over psychosocial matters that concerned the patients or emotional tension in the patients. However, despite the psychosocial impact on the patients resulting from good physical care, senior nurses tended to regard this as simply part of their everyday routine.

> D: 'Looking at your example of how the patient's worries were reduced by the nurse's reassurances and explanations about the physical issues, it happens every day that many patients voice their physical concerns and issues. Hence, it is so common and I won't even think anything of it, and perhaps it is already part of our everyday life and work.'

> (F1)

On the other hand, junior nurses saw the importance of the interplay between physical and psychosocial care.

C: 'As a junior nurse, I found that the nurse in this scenario did quite well in attending to the patient's physical concerns by providing an explanation, reassurance, and support, so she was able to attend to both the patients' physical and psychological needs.'

(F1)

Both senior and junior nurses agreed that, while they are aware of the importance of psychosocial care in cancer wards, they would attend to it only if time permits.

**Intentional psychosocial care.** The nurses commented that they would talk to patients primarily while engaged in providing procedural care; otherwise, they lacked the time to talk to them.

G: 'Usually I spend more time communicating during admission and discharge. Admission is an ideal opportunity . . . to become familiar with the patients.. . . As for discharge, . . . there is a lot to follow up on and explain to the patients. . .. Other than that, you really don't get to communicate with the patients.'

(I)

K: 'During visiting hours, we can talk to the patients and the family and ask the patients what they are eating and the family what they cooked for the patients—and the patients and the family will feel that the nurse cares.'

(F2)

**Unintentional psychosocial care.** Given the grand narrative of the need for oncology nurses to provide psychosocial care to patients, the nurses' self-reflections revealed that despite the lack of time for psychosocial care, they still tried to find different ways to respond to patients' cues and concerns.

C: 'I observed from the patient's facial expression that the patient didn't seem to understand while listening to the doctor's comments—when we have the chance [we will explain things to the patient] and so squeeze some time for that if possible.'

(F1)

J: 'Sometimes we observe that the patient has no visitors; then, we would find out from the patient whether he/she lives alone . . . for future discharge planning.'

(F2)

The key difference between intended and unintended care of patients' psychosocial needs lies in whether the care is planned or unplanned. In both cases, the nurses draw on their repertoire of knowledge. They attend to this secondary matter on their own initiative during the provision of procedural care or when prompted by the cues and concerns of the patients and/ or their families.

## Managing an emotionally challenging environment

**Nurses' experiences of emotional control.** Thinking ahead to potential problems and dealing with them before they occur was noted from the way that the nurses coped in their time-strapped environment. They acted proactively to save time and manage the expected

needs of the patients and their families. This included working to contain the emotions of the patients and/or families, anticipating what they wanted to know, and getting the information to them before they needed to ask. The intention was to build a better relationship and/or to prevent complaints from being raised.

> D: 'I try to anticipate their needs before the patients get frustrated because they do not know what might happen next.'
>
> (F1)
>
> H: 'The patients would like to know what you have done, . . . If they know that you have done something and that you did care for them, actually many conflicts can be settled.'
>
> (F2)
>
> G: 'Empathy is important for nurses. We have feelings and tensions when a patient throws a temper tantrum. We need to be calm, no matter what, and need to see the issue from the patient's perspective. This will help with nurse-patient communication. Often, the patients will wonder about how you can help, or ask how you can understand them, given that the junior nurses are so young.'
>
> (I)

The nurse participants also noted that the stress of illness elicits many emotions in patients and their families, and that these emotions are commonly expressed to nurses. All of the participants said that they found the anger expressed by patients and/or their family members 'difficult' to manage. This emotional management lends support to the emotional labour [38] of nurses. While emotion is considered to be essential for the development of meaningful relationships with patients and motivates the decisions and actions of nurses [39], Henderson [40] asserted that nurses make a choice between emotional engagement and detachment in their emotional caring. Carmack [41] emphasized a need for nurses to strike a balance between detachment and engagement for their well-being.

**Strategies for nurses to control their emotions.** In the findings, nurses described using varying degrees of connection or disconnection to maintain their emotional control and self-efficacy [42].

> B3: 'It is frustrating when patients ask the same questions. Often, I would have already explained things to the patient, but they will still continuously ask the same question. I would indicate that my answer would be the same and step out for a moment, knowing that the patient was looking for a certain response, which was not the one that I gave.'
>
> (F2)
>
> B: 'I think I need to tell them that I am needed by someone else, so that I can leave the room for some time to calm down and for them to quiet down as well.'
>
> (F2)
>
> A: 'I let them ventilate first and talk to them after they settle down. Sometimes after they ventilate, they will apologise.'
>
> (I)

Patients' emotions affect nurse-patient communication, and the nurses advocated allowing both the nurse and the patient to have some physical space to compose themselves. The nurses also commented that while they have theoretical knowledge about empathy and its importance, it would be difficult to put this knowledge into practice given the limited time that they have to spend with patients.

> C: 'If I have a patient who is having a temper tantrum and another who is vomiting blood, I would need to attend to the physiological needs of the second patient first, and let the first patient know that I will be back to talk to him/her.'
>
> (F1)

Nurses used various ways to control the patients' emotions as well as their own, by stepping out and being empathetic. However, it is evident that nurses in the study needed to engage in considerable emotional work and had learnt to regulate or control their emotions based on the context of the situation.

In their responses to a patient's account of an interaction with a nurse, the nurses shared different experiences and values, and held different views on the appropriate choice of words and the approach that the nurse should have taken. Some thought that the nurse was justified in using a stern tone of voice because of the urgency of the situation, while others disagreed, stating that it would affect the nurse-patient relationship.

> H: 'Perhaps it is not so much an emotional issue, as about the [characteristics of the] patient; for those who are sensible, the nurse would behave differently. . .. If we are not firm with our tone, the patient might not get the message.'
>
> (F2)

> F: 'I understand the nurse's concern about the patient's safety . . . she could have used a softer tone . . . but [one] shouldn't put all the blame on the nurse . . . in such a busy environment.'
>
> (F1)

> G: 'I think the tone could be adjusted . . . and the rationale explained for stopping the patient from taking his/her own medications.'
>
> (I)

Some participants espoused the virtues of being patient, listening, and reasoning, while other nurses would seek help from a member of the patient's family to communicate with an emotional patient.

Although the nurses expressed different values and beliefs in the discussion, they all made decisions on the spur of the moment, based on their experiential and theoretical knowledge. This knowledge would grow with continued learning and reflection.

**Lacking control beyond the provision of psychosocial care.** Environment does not simply refer to situations that evoke emotions in nurses when dealing with the physical and psychosocial issues and concerns of cancer patients, but also to the busy work environment. Repeated requests from patients for analgesics may affect nurses emotionally, as they may be unable to fulfil such requests.

B: 'For pain, we have to wait for the doctor to prescribe something; if he/she doesn't come, we can't administer anything for the patient's pain, despite the patient's repeated requests.'

(F2)

G: 'We cannot administer painkillers unless the doctor/ houseman prescribes it. . .. I understand that the patient felt pain . . . but you can really run out of patience after you have explained [the situation to the patient] once or twice, but the patient continues to make such a request.'

(I)

Similarly, when carers are experiencing heightened emotions and stress, delayed responses may create dissatisfaction with nurses.

J: 'Patients would like others to help [them deal] with their emotions, but with too many patients—we might not be able to help them in a timely fashion. Sometimes, a family member might ask you to get a blanket, but you might be distracted by other tasks, so even bringing the blanket might have taken longer [than expected], so the family member might think that the care was not satisfactory—and create some conflict.'

(F2)

Earlier comments from a nurse participant about using the visiting hours to nurture a relationship with the patient and his/her family may reflect an intention to lessen complaints from any delay. While building relationships with patients and their families may help to assuage the patients' grievances, the senior nurses still agreed with the assertion that without the capacity and ability to respond to the needs of many patients, good communication would become an urgent issue in mentoring.

## Mentoring and learning

**Learning through experience.** Experience is important for nurses, along with the need for continuous reflection. Junior nurses might not have good communication skills and also lack an understanding of the patients' health conditions; thus, they might be unable to provide comprehensive answers to the patients' questions.

Seniors provide mentoring to juniors by drawing on their experiences and awareness of situations. Some senior nurses commented that they were still learning about communication and how to meet the needs of patients.

F: 'We are also learning about how best to communicate, and the juniors will have to learn what works in different situations.'

(F1)

H: 'You can't ask the nurses to change their personality or their ways of approaching things. There is a need to nurture the nurses about patient-centered care, or it really is a matter of personality when it comes to communication and commitment to the patients vs clinical governance. Although difficult, value-based care can be learnt and developed.'

(F2)

Nonetheless, having a role model will affect the attitude of fresh graduates towards their work and communication with others.

D2: 'I watched how the seniors communicated with the patients . . . then tried it out with other patients.'

(F1)

G: 'Having senior colleagues as role models will have a great influence on your work attitude, especially in nurse-patient communication in the junior years.'

(I)

However, the point is not to simply observe and follow the senior nurses' ways of doing things.

B3: 'The different personalities, values, and beliefs of the nurses affect their approaches to patients—so while observing how the seniors manage the situations, the juniors eventually have to try out what works and what does not work for them, to develop their own way of approaching the patients in certain situations.'

(F2)

While junior nurses could learn from observing the actions of senior nurses, authenticity of practice will occur only when the junior nurses internalise values and learn to be flexible and creative in response to the context.

A: 'I remember once there was a patient who passed away. His wife and children were crying intensely and refused to let us proceed further. The situation wasn't resolved for two hours. . ., not until I simply touched her shoulder and told her that the children would be with her. . .. I learnt at that moment that nonverbal behaviour could sometimes be a very effective way of calming the patient.'

(I)

The junior nurse learnt from experience that using non-verbal methods can be effective at calming patients, but she did so by also embracing the value of empathy in nurse-patient communication.

## Learning through training

Interactions through communication are complex. While training is valuable, continuous practice and coaching are also important.

F: 'I don't think anyone adjusts to the oncology ward immediately—it takes time. They have many cognitive challenges from patients, and therefore require continuous training. The need for training is not always met in this [fast-paced, stressful, and noisy] environment, and this inadequacy is an added challenge to adjusting to this environment.'

(F1)

J: 'There is not much training in communication . . . but after a training course, there is a need to encourage the juniors to give it a try; if not, they might not.'

(F2)

A: 'I can't rule out the effectiveness of communication training. . .. [But] without a chance to practice them in real clinical encounters, these strategies are easily forgotten.'

(I)

Nurses saw training as involving a lot of theories without much practice. Individual approaches to implementing techniques learnt in training courses need to be taken.

## Discussion

Similar to many others [19, 43], the nurses in this study recognized the importance of attending to psychosocial needs in cancer care. Given the perception of time pressures, however, the findings of this study reinforce Vinckx, Bossuyt [23] insights on nurses' strategies in managing psychosocial care, in that they show that providing psychosocial care came second to attending to the patients' physical demands and physiological changes. Being unaware of the significant interplay between the physical and the psychosocial, and therefore of the psychosocial impact of their physical care on the patients, the nurses in our study indicated that this impact is something that is seen as routine and is taken for granted, especially by senior nurses. However, recognition of this impact is important, and needs to be reinforced for junior nurses. This is because the nurses' sense of identity and their emotional equilibrium could be threatened if they perceive that they have failed to provide the expected psychosocial care [44].

It can be anticipated that cancer patients will experience a great deal of psychosocial distress and may feel vulnerable when faced with the possibility of dying. In a busy oncology environment, the complexity of psychosocial care through nurse-patient communication requires nurses to manage the emotions of the patients and their families arising from anxiety over the patients' health or from complaints/misunderstandings due to a shortage of nurses to attend to the needs of the patients. Some cancer patients could be in denial and express this as anger toward the nurses, which would pose significant challenges to nurse-patient communication [16]. Ineffective communication would undoubtedly also leave patients with feelings of anxiety, frustration, and dissatisfaction [45]. Watts, Botti [46] found that some nurses would avoid facing their own negative emotions towards difficult patients and only attend to the patients' physical needs.

Nurses' perceptions of time constraints are evidently an issue in their prioritizing physical over psychosocial care. That they are expected to emphasise the psychosocial needs of patients in the oncology setting continues to cause nurses to monitor the patients' emotional needs and to provide ad hoc care if possible. Our findings are similar to those of Pehrson, Banerjee [10], and Vinckx, Bossuyt [23] on the proactive and ad hoc strategies that nurses use to manage their time.

As a proactive strategy, it was found that nurses would intentionally talk to patients while providing routine care and follow up on non-verbal cues that they had observed from patients if time permitted. Nurses' actions in this case may reflect the social and professional expectations from patients and nurses themselves.

Nurses often struggle to cope with their own reactions and emotions [47]. The nurses in our study learnt to control their own emotions. They recognized that their personal emotions in response to clinical situations made communicating with patients more difficult. They used

the strategy of stepping back to allow both patient and nurse sufficient space to cool off and resume their talk later, or of being firm in giving consistent information. Nurses may experience their emotions as dissonance, which requires them to suppress frustration. For instance, they would instinctively feel sympathetic to the suffering of the patients and would manage their emotions through detachment in order to care for the patients and achieve emotional harmony [48]. Ultimately, a nurse's choice of words and tone of voice are important, and the use of the right words with patients is pivotal to controlling emotions and conveying empathy [16].

The nurses' struggles with their own and the patient's strong emotions may be better informed by Hochschild [38] notion of presentational forms of emotional labour, (where social rules guide nurses' management of their emotions) and prescriptive forms of emotional labour, where organisational or professional codes of conduct dictate the way nurses manage their emotions. As a whole, emotional labour [38], in the context of the work that nurses do, refers to thoughts and emotions that nurses feel inwardly but cannot express in practice [49]. It is conceptualised as the work that nurses do to manage strong emotions in cancer care when they perceive a misalignment between their inner emotions and those that they are expected to display. This dissonance [48] leads to emotional labour. Working in distressing situations when caring for [cancer patients in particular], makes it necessary for nurses to work on their own emotions [50]. Hence, being compassionate is another form of emotional labour [51]. Along with these few forms, Theodosius [52] argued that it is essential for nurses to engage in emotions as part of the construction of a nurse's identity through connecting with oneself and the patients. Henderson [40] also emphasised that emotional engagement is essential for quality nursing care.

The seeming engagement and disengagement of nurses' emotions with their cancer patients can undoubtedly be better understood through the concept of emotional labour [38]. While our nurses have subscribed to the professional rules to be warm and empathetic and at the same time composed and objective, some have also moved beyond these rules at their own discretion. Understanding the different types of emotional labour required in different situations will be useful for our nurses [12]. Karimi, Leggat [53] suggested that the practise of emotional labour involves the complex ability to contain one's emotions as an internal process while at the same time helping others to explore their emotions as an external process.

During the use of a vignette of a patient's account of her view of nurse-patient communication, which was used to solicit comments from the nurses in the interviews, the majority of the nurses recognized the importance of providing psychosocial care for oncology patients through listening, using appropriate language, and attending to the patients' non-verbal cues. Hochschild [38] suggested that their thoughts may be related to the rules of feeling, which are 'standards used in emotional conversations to determine what is rightly owed and owing in the currency of feeling' (p. 18). Our nurses seem to have engaged in the emotional rules of both presentational and prescriptive forms of labour with compassion.

There were however a couple of nurses who were unaware of how they might have blocked the verbal behaviour of the patients, and the negative impact this might have on their trust and rapport with the patient and, ultimately, on the quality of the care that they could deliver to the patient. For example, the tone of voice and words of one nurse silenced a patient whom she mistakenly thought to be on the verge of taking her own medication from home on top of what was administered by the nurse. Wilkinson [14] talked about facilitating and blocking the verbal behaviors of the patients. Even with prompts and even after having heard the views of her peers, who were more empathetic and sensitive to the patient's needs, the nurse did not seem to aware of her blocking behaviour. In fact, she referred to a characteristic of the patient, which may have had the effect of shifting the blame to the patient [54].

Similarly, while the nurses in this study averted the possibility that patients and their families might voice complaints or frustrations by adopting preventative communication strategies, one nurse's account of providing the patient with information based simply on assumptions from her experience could be regarded as a blocking technique if it prevented the patient from divulging the problem. At the same time, the nurse was also fully aware that the verbal behaviour that she adopted would prevent her from having to cope with the patient's anger and/or complaints. Hence while many proactive strategies used by nurses are seemingly positive, nurses should strive to recognize that each clinical encounter is unique, given that each patient is an individual. It is evident that the nurses made a good effort to handle the strong emotions expressed by patients, but that they also struggled because they did not necessarily have the skills to deal with such situations.

The participants referred to the role played by personality in the communication required for emotional care and in the strategies that they used to deal with the patients' emotions and their own, which led to the notion of emotional intelligence [55]. Hogg, Hanley [12] defined emotional intelligence as 'assessing, demonstrating and controlling emotions and using these skills to modify the emotional states experienced by oneself and others' (p. e1010). However, Snowden, Stenhouse [56] cautioned against drawing a direct correlation between emotional intelligence and an individual's ability to provide sensitive care, as other educative factors such as reflection, training, role modelling, and mentorship are also needed to achieve the ability to provide such care [39]. This understanding aligns with the participants' thoughts. Within the framework of emotional labour and emotional intelligence, nurses and nurse leaders could be encouraged to integrate this understanding along with communication training for oncology nurses. The objective would be to minimize the nurses' surface acting for emotional labour, which would be more likely to result in burnout than in deep acting [57].

Understanding effective nurse-patient communication is not only about developing skills through training and experience over time specifically in oncology settings; rather, the complexity and uncertainty involved in providing this care can leave both nurses and patients struggling with their emotions. This emotional entanglement clearly revealed itself when the oncology nurses in this study commented on their work environment. Already faced with time constraints with an increase in the nurse-patient ratio, nurses also need to deal with repeated requests for medication from patients or family members, which need to be followed up by a busy physician who might not always respond to the nurses who must convey that request. Frustration with the physician may manifest as staff conflict, and could create stress for nurses. Wilkinson [14] considered frustration with staff conflict as a stress predictor of nurses being the high blocker in their communication with patients.

The exceptional level of skills required of nurses to communicate and manage their emotions in their work is evident. The 'tentative' nature of nurse's psychosocial care, even though many would try to follow up, also points to the possibility that they may lack knowledge and skill in communication and in attending to the patient's psychosocial needs. While nurses' communication skills in cancer care would not necessarily improve with experience alone, insufficient training in communication could contribute to emotional stress and burnout. Although all of the participants indicated that they had received some training in communication skills, typically involving theoretical workshops and the sharing of case studies, the majority had not received any follow-up training or reinforcement of their learning in the wards. Those who expressed the view that they lacked the skills to deal with the emotions of patients and needed to be mentored were nurses with primarily of 2–3 years of experience in oncology settings. They commented that it was a challenge for them to apply their theoretical understanding, which often focused on such matters as being respectful, thinking from the patient's perspective, and exploring the emotional stress of patients, rather than on how to do these

things. Communication skills training has long been criticised for deviating from actual practices [58].

Our findings echoed those of Salmon and Young [15], particularly those on our nurses' perceptions of the gap between theory and practice and their continued scepticism about the effectiveness of communication training for complex clinical encounters. Nurses recognized communication as dependent on context and impossible to reduce to a set of theoretical principles or 'golden rules' [59]. Our nurses resorted to experience to guide their practice, reflecting the importance of mentorship in the effective transfer of knowledge. Ultimately, continuous reflective learning for both junior and senior nurses is important for mentoring and coaching in practice. While providing training in communication skills along with clinical supervision [46] is seen as a means to support the consolidation and maintenance of such skills over time and their translation into practice [60], it may also reduce the perception of a lack of time by allowing health professionals to integrate these skills into their daily routines.

Our findings differ from those of Vinckx, Bossuyt [23] in that our nurses described time pressures as giving rise to negative feelings, without mentioning any positive pressures to carry out care more productively. Specifically, the nurses had negative feelings about their care when they felt that they lacked the time to provide care that was considered important.

Working under time pressures often strained nurse-patient communications and interactions with patients. In a time-constrained environment, the individual characteristics (e.g., beliefs and values) of the nurses could influence their view of their priorities [23]. Given the vulnerability of oncology patients, it is important for nurses to be mindful of the interactional and interpersonal aspects of care when working in such an environment. With the busy workload of nurses, creating more time may not be possible; thus, Vinckx, Bossuyt [23] spoke about developing resilience in nurses. In this connection, we propose mindful communication as one way for nurses to cope with the perceived time pressures. Through mindful communication, nurses could learn to develop more awareness of the present moment spent with patients without making any judgements. During the process of attending to, responding to, and perceiving information, nurses can engage in a mindful exchange [61]. Together with training in how to manage negative emotions, identifying the forms of emotional labour in action while preserving the authenticity of their experience [54] through engaging in their emotions will enhance their communication skills.

In a complex healthcare environment, nurses are expected to work with the unexpected and to suit their actions to the context. Our nurses developed their knowledge from their experiences of caring for patients and/or observing their seniors, which became a theoretical guide to their practice. This process resembles Kolb [62] cyclical model of experiential learning. This model posits that the cycle of experiential learning starts from concrete experiences and moves from reflective observation to abstract conceptualization, then closes off from active experimentation with a return to concrete experience. Through reflective observations of their experiences, our junior nurses derived abstract conceptualizations based on their values and beliefs about patient care. Eventually, the senior nurses will expect the junior nurses to actively experiment with their own approaches through observing the cues expressed by different patients. The reflections of some junior nurses revealed that they actively experimented with non-verbal approaches to calm the family members of patients, as the value of empathy is emphasised in nurse-patient communication.

One should also bear in mind the sheer number of factors that can contribute to the complexity of real-life situations. Nurses often need to transcend their knowledge and experiences to deal with the complex and context-dependent needs of their patients [15]. As such, it is important for communication skills training to capture the flexibility and creativity of communication. In effect, what is required is to combine 'first- hand knowledge' from the learners'

own experiences, which is always evolving, with the complementary 'second-hand knowledge', which could possibly be acquired from senior nurses and from communication training, to guide the everyday practices of nurses [63]. Given the ever-growing complex needs of oncology patients and a time-pressured environment, nurses with strong communication skills, good emotional management, and the ability to communicate through mindfulness will play a pivotal role in influencing patient satisfaction, adherence to care plans, and overall clinical outcomes [15].

## Limitations

This study was carried out in only one hospital, which could potentially limit the transferability of the study. However, the views of both junior and senior nurses were obtained, bringing greater variety to the data. While the comments of the senior nurses might have influenced those of the junior nurses, similar findings were revealed in the individual and focus group interviews in which the juniors were active participants. Oncology nurses are busy people, which also limited the number of available participants; hence the small sample in this study. Nevertheless, data saturation was reached in the collecting of data. Therefore, the findings might also have some degree of relevance to other busy oncology settings.

## Conclusion

In a situation of global nursing shortages, understanding and improving the experiences of nurses by providing a better environment and more support is a major issue in healthcare. Our findings contributed to the practical understanding of how oncology nurses provide psychosocial care within the context of the delivery of biomedical services in the face of time constraints and limited resources, where they realistically place a priority on caring for the physical needs of the patients. Working in an emotionally charged environment where cancer nurses must manage the emotions of patients and their own through different forms of emotional labour involving emotional attachment and detachment, it is clear that openness and flexibility are required to accommodate patient-centeredness. Despite continuously learning through experience, mentoring, and self-reflection, nurses employed pro-active strategies to minimize complaints and tensions. It is worth paying attention to the meanings of their reactive and proactive behaviours. Support should be given to integrate communication education, emotional labour, and mindfulness training into clinical practice with continuous mentoring and the nurses' own reflective practices to promote value-based care.

## Supporting information

**S1 Table. Sample stimulus.**
(DOCX)

**S2 Table. An anonymized table of nurses' backgrounds.**
(DOCX)

**S1 File. Interview guide.**
(DOCX)

**S2 File. Interview guide in Chinese.**
(DOCX)

**S1 Fig. Initial thematic map of the qualitative findings.**
(DOCX)

**S2 Fig. Developed thematic map of the qualitative findings.**
(DOCX)

**S3 Fig. Final thematic map of the qualitative findings.**
(DOCX)

## Acknowledgments

The authors would like to thank all of the participants for their time and contributions to the study.

## Author Contributions

**Conceptualization:** E. Angela Chan, F. Y. Wong, Winsome Lam.

**Data curation:** Pak Lik Tsang.

**Formal analysis:** E. Angela Chan, Pak Lik Tsang, Shirley Siu Yin Ching, F. Y. Wong, Winsome Lam.

**Funding acquisition:** E. Angela Chan.

**Investigation:** E. Angela Chan, Pak Lik Tsang, F. Y. Wong, Winsome Lam.

**Methodology:** E. Angela Chan, Winsome Lam.

**Validation:** Shirley Siu Yin Ching, F. Y. Wong.

**Writing – original draft:** E. Angela Chan, Pak Lik Tsang.

**Writing – review & editing:** E. Angela Chan, Pak Lik Tsang, Shirley Siu Yin Ching, F. Y. Wong, Winsome Lam.

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
