## [Decision Letter · Decision Letter 0]

7 Aug 2019

PONE-D-19-16364

Re-examining psychosocial care through nurse-patient communication in busy oncology wards: A qualitative study

PLOS ONE

Dear Dr Chan,

Thank you for submitting your manuscript to PLOS ONE. After careful consideration, we feel that it has merit but does not fully meet PLOS ONE’s publication criteria as it currently stands. Therefore, we invite you to submit a revised version of the manuscript that addresses the points raised during the review process.

For your revision, you may wish to refer to the COREQ guidelines on reporting of qualitative research, as a number of comments from the reviewers highlight this limitation in the manuscript. In particular, please emphasize on how saturation was assessed in this work and elaborate further on the development and finalisation of the thematic map. 

We would appreciate receiving your revised manuscript by Sep 21 2019 11:59PM. To enhance the reproducibility of your results, we recommend that if applicable you deposit your laboratory protocols in protocols.io, where a protocol can be assigned its own identifier (DOI) such that it can be cited independently in the future. For instructions see: http://journals.plos.org/plosone/s/submission-guidelines#loc-laboratory-protocols

We look forward to receiving your revised manuscript.

Kind regards,

Janhavi Ajit Vaingankar

Academic Editor

PLOS ONE

Journal Requirements:

1. Please include a copy of the interview guide used in the study, in both the original language and English, as Supporting Information, or include a citation if it has been published previously.

Reviewers' comments:

Reviewer's Responses to Questions

**Comments to the Author**

1. Is the manuscript technically sound, and do the data support the conclusions?

Reviewer #1: Partly

Reviewer #2: Partly

2. Has the statistical analysis been performed appropriately and rigorously? 

Reviewer #1: N/A

Reviewer #2: No

3. Have the authors made all data underlying the findings in their manuscript fully available?

Reviewer #1: Yes

Reviewer #2: Yes

4. Is the manuscript presented in an intelligible fashion and written in standard English?

Reviewer #1: Yes

Reviewer #2: Yes

5. Review Comments to the Author

Reviewer #1: This is an interesting article. I suggest that the title is reviewed so the focus is nurse-patient communication in oncology wards. There needs to be a clearer focus on skilled communication throughout the essay this should be the focus.

An anonymised table of who the participants are and how long they have worked in oncology should be included.

It isn't clear why focus groups and interviews are both used in the study as nurses can give different accounts according to which method is used.

The focus of the article should be on the nurses response to patients' emotions and their own emotional response. This focus would fit with theoretical work related to 'emotional labour' please see the work of Pam Smith, which is a seminal text on nurses emotional labour.

More information is needed on what findings from the first part of the study are relevant to the current study.

More reflexivity is needed on the data collected from the focus groups and the data from individual interviews.

It isn't clear how the data were coded and how psychosocial care, values and emotions were coded from the data. There needs to be explanations of how comparisons were made between the different types of data. More information is needed on what codes were considered significant and patterned these need to be identified.

A thematic map is referred to again this map needs to be discussed and the process of developing this map further explored. More information is needed on how the final three themes were agreed.

The most interesting part of the data analysis is around nurses' experience of emotional control. It seems that nurses used preventative communication strategies to avert the possibility of complaints or frustrated patients and their families. Once discussing patients' emotions nurses were able to express the strategies they used in communication with those who were distressed in this way. In particular there is a focus on patients expressing anger. The data analysis should focus on the nurses descriptions of connection and disconnection how they facilitated or blocked patients from expressing strong emotion. Please see the work of Wilkinson et al related to barriers to communication. Although the nurses prioritise physical care they also identify their role in going back to communicate with patients who are distressed. There are some expressions where nurses identify their lack of skill in dealing with patient emotions this needs picking up on and relating to what preparation these nurses have had in terms of training as well as years of practice. There is evidence that nurses pass the blame for the situation to others such as the characteristics of the patient or shift the blame for the situation to others.

There is evidence that the nurses struggled with how to handle strong emotion and did not have the skills to deal with this.

The discussion brings out the importance of proactive strategies and this is positive. The discussion should be more focused on the positive strategies but also on the difficulties that nurses experience in handling strong emotions. This later point is not only related to time pressures.

Reviewer #2: This study explores the experiences of nursing communication with respect to emotional concerns based on a series of interviews and focus groups.

The methodology description does not provide the theroretcial framework underpinning the qualitative methodology employed in the study. The description of the conduct of the focus groups is unnecessary and the manuscript would benefit from a description of the questions asked - the supplementary table only includes a copy of the two scenarios used to orientate participants not the interview schedule. Under participants the authors stae that 11 participants is sufficient to reach theretical saturation - how was this establsihed. Was there any check that saturation was reached. Typically saturation is determined empirically - interviews are conducted until no further themes can be identified and this is then checked by conducting a further interviews to confirm. It is uncler whether the two reviewers coding the data independently and how differences were reconciled. Overall there needs to be much greater detail provided for the methodology and rigor checks.

The results : unsure of the context for lines 187-190 - this paragraph does not seem to fit the context of the surrounding text.

The presentation of themes are descriptive rather than a thematic analysis and there are too few quotes or discussion of the complexity of diffrent nursing experiences.

6. PLOS authors have the option to publish the peer review history of their article (what does this mean?). If published, this will include your full peer review and any attached files.

Reviewer #1: Yes: Anne Arber

Reviewer #2: No

---

## [Author Response · Author response to Decision Letter 0]

3 Sep 2019

Responses to reviewers’ commments

Reviewers’ comments Responses

you may wish to refer to the COREQ guidelines Thanks for the suggestion, COREQ was used as a guide. 

Include a copy of the interview guide used in the study, in both the original language and English, as Supporting Information, or include a citation if it has been published previously.

 Copies of the interview guide are included as supporting document. 

Title is reviewed so the focus is nurse-patient communication in oncology wards. There needs to be a clearer focus on skilled communication throughout the essay Thanks for the suggestion. The title has been revised with a focus on nurse-patient communication in oncology wards.

An anonymised table of who the participants are and how long they have worked in oncology should be included. An anonymized table is included.

Reason for why focus groups and interviews are both used in the study as nurses can give different accounts according to which method is used.

 Similar to other researchers, both methods were combined for practical considerations, that is individual interviews were offered to accommodate the busy schedule of the nurse participants, who were unable to attend the focus group (Rees et.al.2003,Taylor 2005).

Focus of the article should be on the nurses response to patients' emotions and their own emotional response, which fit with theoretical work related to 'emotional labour' Thanks for the suggestion, literature on emotional labor and work from Prof. Pam Smith has been reviewed and intergrated into the article.

More information is needed on what findings from the first part of the study are relevant to the current study. More information has been included in the paper. This primarily refers to the use of findings from patients’ perceptions of their nurse-patient communication in the same oncology wards (first part of the study) to stimulate discussion with nurses. After the guide questions, nurses were asked to reflect on the patients’ vignettes and to share their views on nurse-patient communication in the oncology wards.

Please see the work of Wilkinson et al related to barriers to communication. Thanks for the suggestion. Further reading on Wilkinson’s work has been integrated into the discussion. 

Reviewers’ comments Responses

More reflexivity is needed on the data collected from the focus groups and the data from individual interviews. Thanks for the suggestion, reflexivity on the data collected is included. 

How the data were coded and how psychosocial care, values and emotions were coded from the data. There needs to be explanations of how comparisons were made between the different types of data. More information is needed on what codes were considered significant and patterned these need to be identified.

 Thanks for the suggestion. The data process is made more explicit and comparison of different data was provided. Further information is provided on the significant codes and identified patterns. 

A thematic map is referred to again this map needs to be discussed and the process of developing this map further explored. More information is needed on how the final three themes were agreed. 1. The development of the thematic map is included as figures. Information on how the final three themes were agreed upon is included in the data analysis section. 

The methodology description does not provide the theoretical framework underpinning the qualitative methodology employed in the study. Thanks for the comment. A theoretical understanding underpins the use of interviews as a qualitative methodology is included. 

The description of the conduct of the focus groups is unnecessary and the manuscript would benefit from a description of the questions asked Thanks for the suggestion. The description of the focus group is removed. 

The supplementary table only includes a copy of the two scenarios used to orientate participants not the interview schedule. The interview guide is included.

Under participants the authors stated that 11 participants is sufficient to reach theoretical saturation - how was this established? Thanks for the questions. Changes were made for more clarity and check was made for the saturation. 

Unclear whether the two reviewers coding the data independently and how differences were reconciled. Overall there needs to be much greater detail provided for the methodology and rigor checks.

 Thanks for the comments. More clarity was made to show that two reviewers have coded the data independently and for any discrepancies, they would be discussed at the regular meetings. COREQ is used as the guide to check the process. 

Results - lines 187-190 - this paragraph does not seem to fit the context of the surrounding text. The presentation of themes are descriptive rather than a thematic analysis and there are too few quotes or discussion of the complexity of different nursing experiences.

 The paragraph has been removed and more quotes were included for the discussion of the complexity of different nursing experience. 

Rees CE, Ford JE, Sheard CE. Patient information leaflets for prostate cancer: which leaflets should healthcare professionals recommend? Patient Education and Counseling. 2003;49:263-72.

Taylor B. The experiences of overseas nurses working in the NHS: Results of a qualitative study. Diversity in Health and Social Care. 2005;2:17-27.

---

## [Decision Letter · Decision Letter 1]

8 Oct 2019

Nurses' perspectives on their communication with patients in busy oncology wards: A qualitative study

PONE-D-19-16364R1

Dear Dr. Chan,

We are pleased to inform you that your manuscript has been judged scientifically suitable for publication and will be formally accepted for publication once it complies with all outstanding technical requirements.

With kind regards,

Janhavi Ajit Vaingankar

Academic Editor

PLOS ONE

Additional Editor Comments :

Authors have addressed limitations highlighted by the two reviewers. Appropriate reporting guidelines were followed. Additional information pertaining to reflexivity, coding framework, saturation and results are useful.

Reviewers' comments:

Reviewer's Responses to Questions

**Comments to the Author**

1. If the authors have adequately addressed your comments raised in a previous round of review and you feel that this manuscript is now acceptable for publication, you may indicate that here to bypass the “Comments to the Author” section, enter your conflict of interest statement in the “Confidential to Editor” section, and submit your "Accept" recommendation.

Reviewer #1: All comments have been addressed

2. Is the manuscript technically sound, and do the data support the conclusions?

Reviewer #1: Yes

3. Has the statistical analysis been performed appropriately and rigorously? 

Reviewer #1: N/A

4. Have the authors made all data underlying the findings in their manuscript fully available?

Reviewer #1: Yes

5. Is the manuscript presented in an intelligible fashion and written in standard English?

Reviewer #1: Yes

6. Review Comments to the Author

Reviewer #1: The authors have addressed all the points raised by the reviewers in an in-depth manner. The focusing of the article around proactive and defensive communication now works well. There is now a good discussion and integration of the theory of emotional labour.

7. PLOS authors have the option to publish the peer review history of their article (what does this mean?). If published, this will include your full peer review and any attached files.

Reviewer #1: Yes: Anne Arber

---

## [Editor Report · Acceptance letter]

15 Oct 2019

PONE-D-19-16364R1 

Nurses' perspectives on their communication with patients in busy oncology wards: A qualitative study 

Dear Dr. Chan:

I am pleased to inform you that your manuscript has been deemed suitable for publication in PLOS ONE. Congratulations! Your manuscript is now with our production department. 

With kind regards,

on behalf of

Ms Janhavi Ajit Vaingankar 

Academic Editor

PLOS ONE